green chemistry/environmental chemistry/
organic chemistry

bagasse, *p*-TsOH pretreatment, efficient separation, nanocellulose filaments, thermal stability

**Author for correspondence:**
Shuangquan Yao
e-mail: yaoshuangquan@gxu.edu.cn

†Present address: School of Light Industrial and Food Engineering, Guangxi University, Nanning, 530004, PR China.

This article has been edited by the Royal Society of Chemistry, including the commissioning, peer review process and editorial aspects up to the point of acceptance.

# Effect of *p*-TsOH pretreatment on separation of bagasse components and preparation of nanocellulose filaments

Chengqi Feng[1,2], Juan Du[1,2], Shuai Wei[1,2],
Chengrong Qin[1,2], Chen Liang[1,2]
and Shuangquan Yao[1,2,†]

[1]School of Light Industrial and Food Engineering, Guangxi University, Nanning 530004, People's Republic of China
[2]Guangxi Key Laboratory of Clean Pulp and Papermaking and Pollution Control, Nanning 530004, People's Republic of China

SY, 0000-0003-4982-998X

The efficient separation of bagasse components was achieved by *p*-toluenesulfonic acid (*p*-TsOH) pretreatment. The effects of *p*-TsOH dosage, reaction temperature and reaction time on cellulose, hemicellulose and lignin contents were studied. Eighty-five per cent of lignin was dissolved, whereas the cellulose loss was minimal (less than 8.1%). Cellulose-rich water-insoluble residual solids were obtained. The degree of polymerization of cellulose decreased slightly, but the crystallinity index (CrI) increased from 52.0% to 68.1%. It indicated that the highly efficient delignification of bagasse was achieved by *p*-TsOH pretreatment. The nanocellulose filaments (CNFs) were produced by the treated samples. The physico-chemical properties of CNFs were characterized by transmission electron microscopy and thermogravimetric analysis. The results show that the CNFs have smaller average size and higher thermal stability. It provides a new method for CNFs.

## 1. Introduction

The development trend of woody biomass is to realize the high-value utilization of the whole component [1,2]. The prerequisite is efficient clean separation of the main components. As one of the major components of lignocellulosic biomass, cellulose is a

highly abundant, renewable and biodegradable biopolymer [1,3] and has applications in various areas [4–6]. Nanocellulose is a typical example of high-value utilization of cellulose [7]. As one of the main types of nanocellulose, nanocellulose filaments (CNFs) have low density, high aspect ratio, high mechanical strength, good hydrophobicity, good biocompatibility and biodegradability [8]. The preparation and application of CNFs have attracted wide attention in academia and industry [9,10].

At present, CNFs are mainly prepared by pretreatment and mechanical treatment. Most of the lignin and hemicellulose from lignocellulosic biomass are removed during pretreatment and the remaining solid, rich in cellulose, is obtained. This is due to the fact that the dispersion of cellulose during mechanical treatment is reduced by the presence of a large amount of lignin [11]. To solve this problem, various pretreatment methods have been studied, including alkaline [12], dilute acid [13], sulfite pulping [14], TEMPO oxidation [7], enzyme [13] and solvent processes [15,16]. Appropriate acid or alkali treatment can not only dissolve lignin but also remove impurities such as hemicellulose and pectin from cellulose. However, excessive acid or alkali treatment may also lead to the degradation of cellulose. The sulfite pulping process requires operating at high temperatures, using high-pressure vessels and requiring expensive chemical recovery. Different kinds of enzymes are used in combination in the enzymes pretreatment, and the reaction conditions need to be strictly controlled. Therefore, the low-temperature and rapid delignification with simplistic recovery of chemicals needs to be developed. Chen *et al.* [15] found that 90% of poplar wood lignin was dissolved at 80°C in 20 min by *p*-toluenesulfonic acid (*p*-TsOH) pretreatment. Cellulose, lignin and hemicellulose from lignocellulosic materials were separated efficiently by *p*-TsOH pretreatment. It has the advantages of low reaction temperature and short reaction time [16].

Bagasse as a waste of agriculture and forestry can be widely used in the preparation of nanocellulose [17,18]. In this paper, the separation of bagasse components by *p*-TsOH pretreatment was investigated. CNFs were produced from the pretreated bagasse. The effects of *p*-TsOH dosage, reaction temperature, and reaction time on cellulose, hemicellulose and lignin contents were studied. The degree of polymerization (DP) of cellulose and the crystallinity index (CrI) of treated samples was analysed. The physico-chemical properties of CNFs were characterized by transmission electron microscopy (TEM) and thermogravimetric analysis (TGA). The results provide theoretical support for the preparation and application of CNFs.

# 2. Material and methods

## 2.1. Materials

The bagasse was obtained from Guangxi, China. The chemical composition of untreated and treated bagasse samples was analysed by NREL method. The detail of the method was according to Huang *et al.* [19]. The content of cellulose, hemicellulose and lignin in raw material was 46.98%, 20.84% and 23.21%, respectively. *p*-TsOH (99.5%, analytical grade) was obtained from Sigma-Aldrich (St Louis, MO, USA). All the assay reagents were obtained from Aladdin (Shanghai, China).

## 2.2. *p*-TsOH pretreatment

The efficient separation of bagasse components was achieved by *p*-TsOH pretreatment. The details of *p*-TsOH pretreatment of bagasse raw materials were as follows. The *p*-TsOH solution was added to a flask and heated. Bagasse powder (100–180 mesh) was manually fed into the flask at a solid to liquid ratio of 20:1. The fibre suspension was constantly mixed by magnetic stirring at 300 r.p.m. At the end of the reaction, 100 ml of deionized water was added to terminate the reaction. The hydrolysate was separated by vacuum filtration, and the filter residue was washed to neutral. The solid residue was collected, and the lignin was extracted from the reaction solution. The *p*-TsOH was recovered after the reaction.

Bleached bagasse was obtained by cooking and bleaching. The basic method and process were described by Yao *et al.* [20].

## 2.3. Characterization of bagasse samples

The bagasse samples were dissolved by 0.5 M cuprammonium ethylenediamine solution. The DP of cellulose was measured by SI Analytics capillary viscometer (SCHOTT, Jena, Germany). The solution

was passed through the capillary viscometer after the fibre was completely dissolved. The time was recorded as a good approximation of the cellulose DP, as it excludes the effects of hemicellulose polymers and allows a fair comparison of the effects of various treatments. The DP of bagasse samples were calculated [21].

The CrI of bagasse samples was determined by X-ray diffractometer (XRD) (MiniFlex 600, Rigaku Co., Japan) at 30 kV and 10 mA with Cu radiation (1.54 Å). The pattern was recorded from 5° to 50° with a step size of 0.02°. The CrI of samples were calculated as previously described by Ge *et al.* [22].

## 2.4. Preparation of nanocellulose filaments

In fact, most of the lignin and hemicellulose from bagasse were removed by *p*-TsOH pretreatment. But the cellulose was also damaged in the process, and some lignin remains in the remaining raw material. Their impact on the preparation of CNFs was evaluated. CNFs were prepared from the treated samples using published methods [23]. A suspension of each sample was prepared at a dosage of 0.8% and stirred for 24 h. The solution was homogenized in a high-pressure homogenizer (M-110EH-30, The Netherlands). Homogenization was first performed in a chamber having a pore diameter of 200 µm at a homogenizing pressure of 350 bars, where the number of sample passes was five. Then, homogenization was performed using a chamber having a pore diameter of 87 µm. The homogenizing pressure was 1500 bar, and the number of sample passes was 10. The high-pressure-homogenized sample was placed in a refrigerator at 4°C for subsequent experiments. The control CNFs were prepared by bleaching bagasse pulp. The basic method and process were described by Yang *et al.* [24].

## 2.5. Physico-chemical properties of nanocellulose filaments

The physical dimensions of the samples were observed by TEM (Hitachi HT-7700, Japan) [25]. The operating voltage was 100 kV. The nanofibre dimensions were calculated using Nano Measurer software. The TEM spectra of various samples were obtained. The particle size distribution and average diameter of various samples were calculated.

The thermal stability of the fibre was analysed by TGA (STA 449 F5, NETZSCH, Germany) [22]. First, the fibre sample was dried for 6 h and passed through a freeze dryer (Labconco, Kansas City, MO, USA). Approximately 10 mg of each sample was analysed. The heating rate was 10°C min$^{-1}$. The pyrolysis temperature range was 30–600°C. The nitrogen flow rate was set to 20 ml min$^{-1}$ to ensure that the sample was pyrolysed in a nitrogen environment. The thermal degradation spectrum of the sample and its derivative curve were obtained, and the thermal stability of the various components was evaluated.

# 3. Results and discussion

## 3.1. Effect of *p*-TsOH dosage on bagasse components

Eighty-five per cent of birch lignin was dissolved by *p*-TsOH, and water-insoluble solids rich in cellulose were obtained [16]. The separation of bagasse components by *p*-TsOH pretreatment was studied. The effect of *p*-TsOH dosage on the cellulose, hemicellulose and lignin contents was analysed. The *p*-TsOH dosage was varied from 40% to 90%. The pretreatment temperature and time were 90°C and 30 min, respectively. The results are shown in figure 1.

With the increase of *p*-TsOH dosage, the effect of lignin removal was significant. The content of lignin decreased from 21.33% to 3.17% with *p*-TsOH dosage increased from 40% to 80%. Moreover, the content of hemicellulose decreased from 19.81% to 6.85%. However, the dissolution of cellulose was inhibited. The content of cellulose decreased slightly, from 46.69% to 42.86%. The reason is that *p*-TsOH is a strong organic acid, the α-carbon atom in the lignin was de-substituted, causing the lignin to dissociate into fragments and ultimately dissolve. The α-carbon atom, from which the substituent was removed, formed a positive carbon ion (a strong electrophile) and reacted easily with sulfonic acid (a nucleophile), resulting in sulfonation of the α-carbon atom. In the acidic environment, the glycosidic linkages in the hemicellulose were destroyed, and the hemicellulose was hydrolysed into monosaccharides, which were eluted from raw material. Similarly, cellulose degraded into glucose in the reaction, but the degradation was slower and less extensive. The reason is that the crystalline

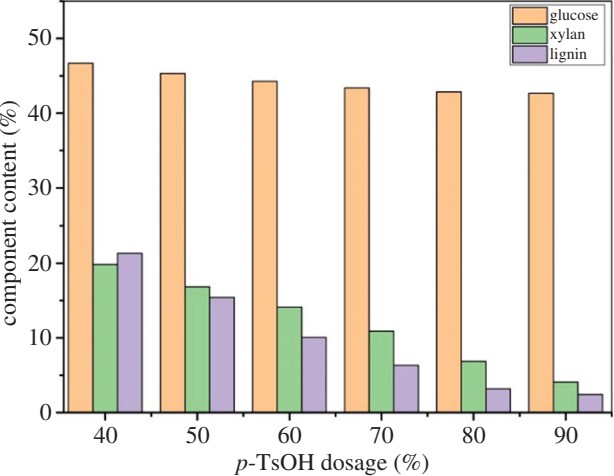

**Figure 1.** Effect of *p*-TsOH dosage on the main component content of bagasse during *p*-TsOH pretreatment.

structure of cellulose was difficult to destroy by *p*-TsOH at a lower temperature. Only a small amount of amorphous structure was degraded, and it dissolved into monosaccharides. When the dosage of *p*-TsOH exceeded 80%, the degradation rate decreased from 3.17% to 2.42% owing to the condensation reaction of the nucleophilic C1 and C6 with the benzene ring in the lignin during lignin dissolution [26]. The condensed lignin fragments reattached to the surfaces of the solids, hindering further dissolution of the lignin and reducing the dissolution rate [27,28]. The content of hemicellulose decreased further, from 6.85% to 4.08%. In the acidic environment, the cellulose content continued to decrease slightly, going from 42.86% to 42.70%. High-performance CNFs need to be produced by *p*-TsOH pretreatment. This means that the lignin and hemicellulose content in the pretreatment samples should be reduced as much as possible, while the cellulose should be protected during the process. The results showed that the contents of lignin and hemicellulose were significantly reduced with the increase of *p*-TsOH dosage. The dosage of *p*-TsOH concentration had little effect on cellulose content. The reduction of lignin and hemicellulose was effective when the dosage of *p*-TsOH exceeds 80%. It also causes waste of chemicals and higher disposal costs. Therefore, 80% *p*-TsOH is the optimal condition for the CNFs production.

## 3.2. Effect of temperature on bagasse components

High temperature was conducive to the dissolution of lignin [29]. However, the dissolution and degradation of cellulose and hemicellulose were intensified with the increase of reaction temperature [30]. The effect of temperature on the main component content of bagasse was analysed. The temperature of the reaction varies from 50°C to 100°C. The other reaction conditions were *p*-TsOH dosage was 80% and the pretreatment time was 30 min. The results are shown in figure 2.

The lignin dissolved rapidly at reaction temperatures of 50°C–80°C, and the content of lignin decreased from 21.17% to 2.21%. In fact, cellulose was less damaged at this temperature range. The content of cellulose decreased from 46.39% to 42.81%. Similar to the change trend of lignin, the dissolution of hemicellulose was promoted. The content of hemicellulose decreased from 19.09% to 4.34%. The results showed that the dissolution of lignin and hemicellulose was improved by increasing the pretreatment temperature. However, similar to the effect of *p*-TsOH dosage on the dissolution of glucan, the dissolution of cellulose was not affected by temperature [15]. A small amount of amorphous structure in the cellulose was removed by acid degradation [31]. As the temperature increased further, the lignin content decreased from 2.21% to 1.93%, but the rate of dissolution decreased as it degraded into smaller fragments [32]. The cellulose crystallization zone was slightly degraded, and the content of cellulose continued to decrease, from 42.81% to 42.70%, although the dissolution rate decreased. The content of hemicellulose decreased from 4.34% to 3.55%. This means that there is no significant effect on the dissolution of hemicellulose when the temperature exceeds 80°C. Thus, the optimal temperature of *p*-TsOH pretreatment was 80°C. The process of dissolving lignin with a solvent or chemical often requires high-temperature treatment. This is to prevent lignin from being precipitated at low temperatures. The pulp thickening and equipment

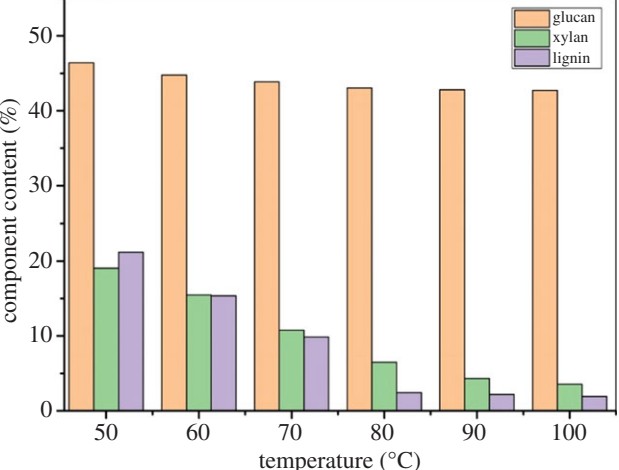

**Figure 2.** Effect of temperature on the main component content of bagasse during *p*-TsOH pretreatment.

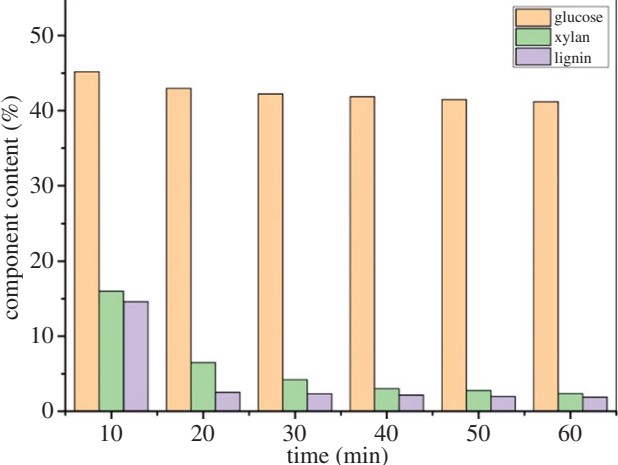

**Figure 3.** Effect of time on the main component content of bagasse during *p*-TsOH pretreatment.

fouling were inhibited. Therefore, the removal of lignin by *p*-TsOH pretreatment at low temperature has a great advantage. In addition, the effects of temperature and addition of *p*-TsOH dosage on separation of bagasse components were compared. This result shows that the *p*-TsOH dosage had a more significant effect on the dissolution of lignin.

## 3.3. Effect of time on bagasse components

In general, the reaction time of separation of lignocellulosic components by organic acid pretreatment was 4 h [33]. The adverse effect of lignin removal on the physico-chemical structure of biomass increased with the extension of reaction time. The development trend of lignin removal is short time and high efficiency. The effect of time on the main component content of bagasse was analysed. The pretreatment time was 10, 20, 30, 40, 50 and 60 min, respectively. The other reaction conditions were *p*-TsOH dosage was 80% and the reaction temperature was 80°C. The results are shown in figure 3.

Compared with *p*-TsOH dosage and reaction temperature, the time had more significant effect on the dissolution of lignin. The dissolution of lignin was significantly promoted within 20 min in figure 3. The content of lignin decreased from 23.21% to 2.53%. The dissolution of hemicellulose was also promoted. The content of hemicellulose decreased from 20.84% to 6.49%. Unlike lignin and hemicellulose, the dissolution of cellulose was less affected by reaction time. The content of cellulose decreased from 46.98% to 42.96%. In fact, as a non-oxidizing strong organic acid, cellulose was not oxidized and degraded in *p*-TsOH pretreatment. But lignin and hemicellulose were extracted efficiently. The

**Table 1.** Effects of different methods on cellulose properties.

| samples | DP | CrI (%) |
|---|---|---|
| bagasse | 3008 | 52.01 |
| bleached pulp | 1200 | 58.41 |
| p-TsOH pretreatment | 2756 | 69.67 |

dissolution rate of lignin decreases gradually as acid-soluble lignin was continuously extracted. Thus, the dissolution rate of lignin and hemicellulose decreases gradually as the reaction time was extended further. The contents of lignin and hemicellulose were 2.33% and 4.23%, respectively, at 30 min. The content of lignin and hemicellulose remained unchanged after 30 min. The changes of lignin, hemicellulose and cellulose contents were compared between 20 and 30 min. The results showed that the improvement of lignin removal could not be achieved by prolonging the pretreatment time. Therefore, the optimal pretreatment time was 20 min.

The optimal conditions for separation of bagasse components by p-TsOH pretreatment were p-TsOH dosage of 80%, pretreatment temperature of 80°C, and time of 20 min. Eighty-five per cent of the lignin and 70% of the hemicellulose were solubilized under optimal conditions. The mass yield was approximately 53%. The cellulose losses were minimal (8.1% or less). Thus, cellulose-rich water-insoluble residual solids were obtained. This indicates that the p-TsOH pretreatment has potential application in the preparation of CNFs.

## 3.4. Analysis of cellulose properties

The purpose of effective lignin removal is to remove lignin while inhibiting the destruction of cellulose structure. The properties of cellulose are mainly determined by the DP and the CrI. The effects of different methods on the DP and CrI were analysed. The results are shown in table 1.

Most of the bleaching agents used in bagasse bleaching are oxidizing. The residual lignin in the pulp is removed using the oxidability of bleaching agents. However, the degradation of carbohydrates (cellulose) in the bleaching process is due to the poor oxidation selectivity of bleaching agents. This resulted in a significant decrease in the DP of the bleached pulp. While lignin was effectively removed, the structural damage of cellulose was inhibited in p-TsOH pretreatment. This is due to the fact that p-TsOH is a strong non-oxidizing acid. The damage to the crystalline and amorphous areas of cellulose was reduced. Consequently, the molecular weight of cellulose drops slightly. The DP decreased from 3008 to 2756, a decrease of only 8.1%, indicating that there was less degradation of the cellulose.

Cellulose, hemicellulose and lignin are the three major components of lignocellulosic biomass. In contrast to hemicellulose and lignin, which exist in an amorphous form in nature, cellulose easily forms a crystalline structure owing to hydrogen bonding and van der Waals forces within and between its molecular chains. Thus, the crystalline composition and structure of cellulose can be reflected in changes in the CrI [34–36]. The CrI of the untreated and treated bagasse samples are shown in table 1.

The result showed that cellulose was present in the form of cellulose I, but not cellulose II, as the main peak did not exhibit an intensity doublet. Thus, the crystalline form of cellulose was not destroyed under different pretreatment. The CrI of bagasse bleached pulp increased from 52.01% to 58.41%. This is due to the large removal of the amorphous component. The cooking and bleaching had a significant effect on the amorphous aggregation state of bagasse. This was attributed to hemicellulose being largely removed in the cooking and bleaching reaction, which represented amorphous components. Thus, the relative content of crystalline cellulose in bleached pulp was increased. In fact, part of the cellulose structure was destroyed by the strong oxidation of the bleaching agents. By contrast, the p-TsOH pretreatment had little effect on the crystalline region of the cellulose. Therefore, the CrI of the p-TsOH-treated sample was higher than that of the bleached pulp. Similar results were reported in a previous study [37]. Accordingly, the results showed that p-TsOH treatment under mild reaction conditions caused only minor changes in the cellulose structure. It indicated that the p-TsOH pretreatment is a highly efficient component separation method.

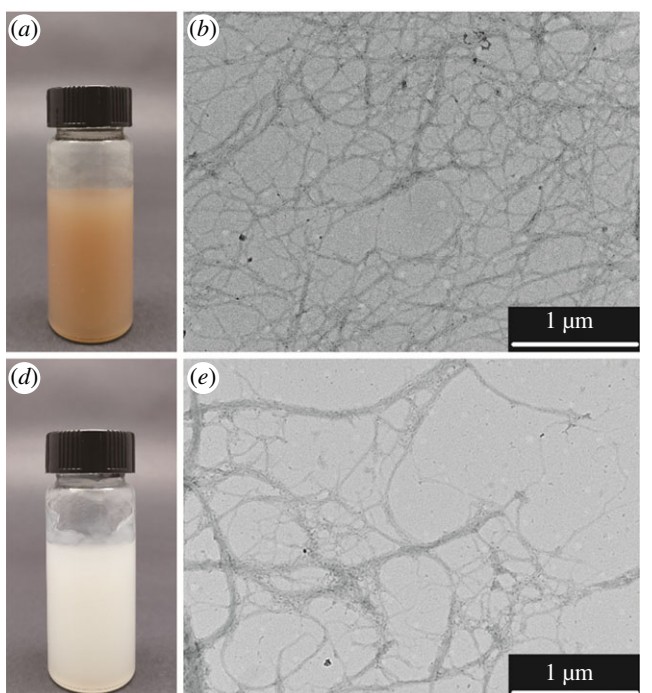

**Figure 4.** Morphology and size of CNFs and the control CNFs. The pictures of (*a*) CNFs and (*d*) the control CNFs; TEM of (*b*) CNFs (*b*) and (*e*) the control CNFs; diameter distribution of (*c*) LCNFs and (*f*) the control CNFs.

## 3.5. Morphology and size analysis of nanocellulose filaments

CNFs are more environmentally sustainable by reducing bleaching reactions, which also leads to higher yields [38]. Moreover, CNFs can be directly produced from raw woody biomass by low-cost and environmentally friendly pretreatment, and no commercial pulps were used [16]. The physico-chemical properties of CNFs and the control CNFs were compared. The morphology and size of CNFs and the control CNFs were analysed by TEM. The results are shown in figure 4.

In figure 4, there were obvious different colour in the actual pictures of CNF (figure 4*a*) and lignocellulose nanofibres (LCNF) (figure 4*d*), because of the different content of lignin. Compared to the control CNFs, the filaments were wound together to form a more compact network in CNFs (figure 4*b,e*). In fact, the cell wall of bagasse was destroyed and the hydrogen bonds between the cellulose fibres were weakened by *p*-TsOH pretreatment. The microfibres were exposed and more easily fibrillated using the final mechanical method, which is consistent with published results [39]. Much of the lignin and hemicellulose in bagasse was removed during cooking. Residual lignin was oxidized and degraded during bleaching. However, a small amount of cellulose was oxidized and degraded in the bleaching reaction. The amorphous area of cellulose in the control CNFs was greatly reduced. The entanglement between the filaments was weakened. From the results of morphology analysis, CNFs have higher toughness. The properties of elasticity, impact resistance and bending resistance are better in composite materials.

The average diameter of CNFs was 17.8 nm, which was smaller than that of the control CNFs (approx. 24 nm), although both were nanoscale fibres in figure 4*c,f*. The diameter distribution of CNFs was more concentrated, and it had a higher aspect ratio than the control CNFs. The CNFs had a uniform diameter and were slightly agglomerated because of hydrogen bonding on the surfaces of the nanofibres [40]. They exhibited better dispersibility and better dispersion than the control CNFs. Nano-spherical lignin particles, which were formed by mechanical action and thermal softening, appeared in the control CNFs [26,41]. The nano-lignin particles were evenly distributed on the surface of the cellulose and in the fibre network structure [42]. The particles on the surfaces of the fibres interfered with hydrogen bonding between the fibres, which improved the dispersibility of the nanocellulose. This indicates that the CNFs have excellent prospects in the application of nanocellulose materials.

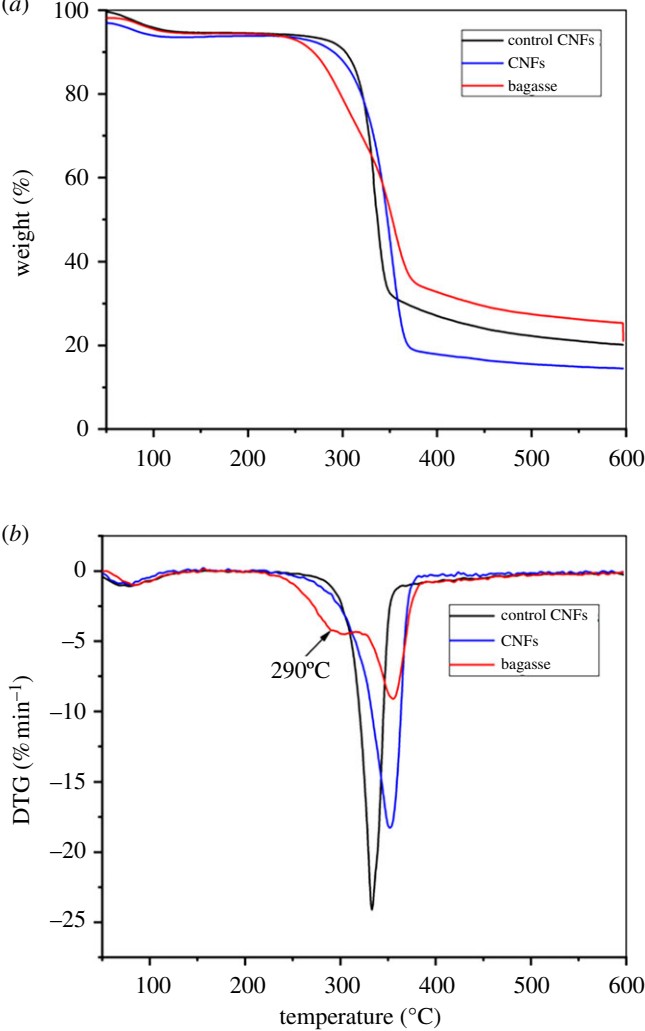

**Figure 5.** Thermal analysis of CNFs (*a*) and the control CNFs (*b*).

**Table 2.** Thermal properties of CNFs and the control CNFs.

| samples | $T_{on}$ (°C) | $T_{max}$ (°C) | MWL (% min$^{-1}$) |
|---|---|---|---|
| bagasse | 300 | 374 | 9.11 |
| CNFs | 322 | 365 | 18.27 |
| LCNFs | 327 | 338 | 24.12 |

## 3.6. Thermal analysis of nanocellulose filaments

One of the characteristics of nanocellulose is its high thermal stability. The above research shows that CNFs have a tighter network structure and a smaller average diameter. The effects of these changes on the thermal stability of CNFs were investigated. The thermal behaviour of CNFs and the control CNFs was analysed by TGA. The results are shown in figure 5.

The results of the TGA of CNFs, the control CNFs and bagasse are shown in figure 5*a*. The degradation onset temperatures ($T_{on}$) and temperatures of maximum decomposition ($T_{max}$) of the samples are listed in table 2. The $T_{on}$ value of bagasse was 300°C. The $T_{on}$ value of CNFs and the control CNFs was 27°C and 22°C higher than those of bagasse, respectively. The higher $T_{on}$ values of CNFs and the control CNFs reflect the removal of non-cellulose components, which have lower thermal stability, from these samples. This difference is due to the embedding of hemicellulose and

lignin in the cellulose fibre bundles in bagasse fibres [43,44]. Compared with the control CNFs, CNFs have a higher $T_{on}$ value. This was due to the higher CrI of CNFs. The $T_{max}$ value of CNFs was 36°C lower than that of bagasse. A possible explanation is that the lignin structure was severely damaged during the p-TsOH pretreatment. Small molecules of lignin were deposited and repolymerized on the surface of cellulose. Pseudo-lignin macromolecules were formed. The pyrolysis temperature of pseudo-lignin decreases because its chemical binding force was lower than that of natural lignin [45]. The $T_{max}$ value of the CNFs was 27°C lower than that of the control CNFs. However, the total weight loss value (TWL) of CNFs was 60.53%, which is lower than that of the control CNFs (66.86%). This difference can be attributed to the removal of residual lignin during the bleaching process, which caused the $T_{max}$ value of the control CNFs to increase. Lignin–cellulose complexes can be formed during the production of nanocellulose through interactions between the remaining lignin and cellulose. Therefore, the lignin can protect the cellulose during thermal degradation, increasing the cellulose degradation temperature and reducing the fibre loss.

The derivative thermogravimetric (DTG) curve of the bagasse shows a major decomposition peak at 355°C and a shoulder peak at 290°C (figure 5b), which was attributed to the degradation of α-cellulose and hemicellulose, respectively. The shoulder peak was absent from the curve of the CNFs and the control CNFs, as hemicellulose was removed during the preceding treatment [46]. Compared with bagasse, the maximum weight loss rate (MWL) of the control CNFs increased to 18.27% min⁻¹. This demonstrates the high thermal stability of nanocellulose. In particular, the MWL of CNFs was significantly increased to 24.12% min⁻¹. The results show that the retention of lignin in nanocellulose was very important for the improvement of thermal stability. The good thermal stability of the CNFs results mainly from the very small number of chemical reagents used in the pretreatment process, as a result of which the cellulose fibres were not seriously damaged.

## 4. Conclusion

A highly effective delignification was discovered. The degradation of cellulose was inhibited while most lignin was removed during p-TsOH pretreatment. In addition, the DP of treated bagasse decreased slightly, while the CrI was higher. The application prospect of this new pretreatment for preparation of CNFs was discussed. CNFs were prepared from treated bagasse. Compared with the control CNFs, CNFs have higher toughness, smaller average size and higher thermal stability. The results open a new door for the preparation of CNFs.

Data accessibility. The datasets supporting this article have been uploaded as part of the manuscript and electronic supplementary material.
Authors' contributions. Conceptualization: S.Y.; methodology: C.F.; resources: J.D. and S.W.; writing—original draft preparation: C.Q.; data curation: C.L.; writing—review and editing: S.W.; project administration: S.Y.
Competing interests. The authors declare no conflict of interest. The funders had no role in the design of the study; in the collection, analyses or interpretation of data; in the writing of the manuscript; or in the decision to publish the results.
Funding. This project was sponsored by the National Natural Science Foundation of China (grant nos. 21968004 and 31760192).

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
