## [Reviewer comments · Royal Society Open Science]

Review History

RSOS-200967.R0 (Original submission)

Review form: Reviewer 1 (Ankush Biradar)

Is the manuscript scientifically sound in its present form?

No

Are the interpretations and conclusions justified by the results?

No

Is the language acceptable?

No

Do you have any ethical concerns with this paper?

No

Have you any concerns about statistical analyses in this paper?

No

Recommendation?

Major revision is needed (please make suggestions in comments)

Comments to the Author(s)

The manuscript entitled "Effect of p-TsOH pretreatment on the bagasse delignification and preparation of lignin-containing nanocellulose filaments" by Yao et al. described the use of pTsOH in depolymerization of bagasse. The topic is interesting and can be useful for material chemists. The title says that delignification, but actually it appears to me that depolymerization reaction. Also, the results presented are not clear to some reproduced at other places. Also, need to rewrite many parts of the manuscript. Need to explain what happened to lignin. and many more

Review form: Reviewer 2**Is the manuscript scientifically sound in its present form?**

Yes

Are the interpretations and conclusions justified by the results?

Yes

Is the language acceptable?

No

Do you have any ethical concerns with this paper?

No

Have you any concerns about statistical analyses in this paper?

No

Recommendation?

Major revision is needed (please make suggestions in comments)

Comments to the Author(s)

In this manuscript, the lignin-containing nanocellulose filaments (LCNFs) was prepared with bagasse which pretreated by p-toluenesulfonic acid (p-TsOH), then CNFs and LCNFs were compared. The study shows that this is indeed a novel and extremely efficient pretreatment method, the LCNFs has higher toughness, smaller average size and higher thermal stability. This work is interesting and inspiring to the field of guiding significance for LCNFs preparation. The followings are the suggestions,

1 In this manuscript, were LCNFs and CNFs prepared in the same way?

2 It is suggested that Fig. 4. and Fig. 5. should be combined together to facilitate the reader's observation.

3 It is recommended to unify the font, such as "°C" in the first paragraph of Section 4.2.

4 Please specify the reason why 80% p-toluenesulfonic acid (p-TsOH) is the optimal condition for the LCNFs production.

5 Are the three factors, the concentration of p-TsOH, temperature and time, worked independently? Please specify the reason why they were studied separately.

6 The images of the LCNFs and CNFs were highly recommended to be included in the manuscript.

Decision letter (RSOS-200967.R0)

Dear Dr Yao:

Title: Effect of p-TsOH pretreatment on the bagasse delignification and preparation of lignin-containing nanocellulose filaments
Manuscript ID: RSOS-200967

The editor assigned to your manuscript has now received comments from reviewers. We would like you to revise your paper in accordance with the referee and Subject Editor suggestions which can be found below (not including confidential reports to the Editor). Please note this decision does not guarantee eventual acceptance.

Please submit your revised paper before 08-Aug-2020. Please note that the revision deadline will expire at 00.00am on this date. If we do not hear from you within this time then it will be assumed that the paper has been withdrawn. In exceptional circumstances, extensions may be possible if agreed with the Editorial Office in advance. We do not allow multiple rounds of revision so we urge you to make every effort to fully address all of the comments at this stage. If deemed necessary by the Editors, your manuscript will be sent back to one or more of the original reviewers for assessment. If the original reviewers are not available we may invite new reviewers.

On behalf of the Subject Editor Professor Anthony Stace and the Associate Editor Dr Dattatray Late.

RSC Associate Editor:
Comments to the Author:
(There are no comments.)

RSC Subject Editor:
Comments to the Author:
(There are no comments.)

Reviewers' Comments to Author:
Reviewer: 1

Comments to the Author(s)
The manuscript entitled "Effect of p-TsOH pretreatment on the bagasse delignification and preparation of lignin-containing nanocellulose filaments" by Yao et al described the use of pTsOH in depolymerization of bagasse. The topic is interesting and can be useful for material chemists. The title says that delignification, but actually it appears to me that depolymerization reaction. Also, the results presented are not clear to some reproduced at other places. Also, need to rewrite many parts of the manuscript. Need to explain what happened to lignin. and many more

Reviewer: 2

Comments to the Author(s)
In this manuscript, the lignin-containing nanocellulose filaments (LCNFs) was prepared with bagasse which pretreated by p-toluenesulfonic acid (p-TsOH), then CNFs and LCNFs were compared. The study shows that this is indeed a novel and extremely efficient pretreatment method, the LCNFs has higher toughness, smaller average size and higher thermal stability. This work is interesting and inspiring to the field of guiding significance for LCNFs preparation. The followings are the suggestions,
1 In this manuscript, were LCNFs and CNFs prepared in the same way?
2 It is suggested that Fig. 4. and Fig. 5. should be combined together to facilitate the reader's observation.
3 It is recommended to unify the font, such as "°C" in the first paragraph of Section 4.2.
4 Please specify the reason why 80% p-toluenesulfonic acid (p-TsOH) is the optimal condition for the LCNFs production.
5 Are the three factors, the concentration of p-TsOH, temperature and time, worked independently? Please specify the reason why they were studied separately.
6 The images of the LCNFs and CNFs were highly recommended to be included in the manuscript.

Author's Response to Decision Letter for (RSOS-200967.R0)

See Appendix A.

RSOS-200967.R1 (Revision)

Review form: Reviewer 2

Is the manuscript scientifically sound in its present form?

Yes

Are the interpretations and conclusions justified by the results?

Yes

Is the language acceptable?

Yes

Do you have any ethical concerns with this paper?

No

Have you any concerns about statistical analyses in this paper?

No

Recommendation?

Accept as is

Comments to the Author(s)

This manuscript is suitable for publication

Decision letter (RSOS-200967.R1)

Dear Dr Yao:

Title: Effect of p-TsOH pretreatment on separation of bagasse components and preparation of nanocellulose filaments

Manuscript ID: RSOS-200967.R1

It is a pleasure to accept your manuscript in its current form for publication in Royal Society Open Science. The chemistry content of Royal Society Open Science is published in collaboration with the Royal Society of Chemistry.

On behalf of the Subject Editor Professor Anthony Stace and the Associate Editor Dr Dattatray Late.

RSC Associate Editor:
Comments to the Author:
Accept

RSC Subject Editor:
Comments to the Author:
(There are no comments.)

Reviewer(s)' Comments to Author:
Reviewer: 2

Comments to the Author(s)
This manuscript is suitable for publication

Appendix A

Dear Reviewers,

Thank you for your letter and for the comments concerning our manuscript entitled “Effect of *p*-TsOH pretreatment on the bagasse delignification and preparation of lignin-containing nanocellulose filaments”. We have studied your comments carefully and have made corrections which we hope could meet your requirements. All changes have been highlighted in the revised version (red highlighting).

Questions you put forward are explained as follows:

Reviewer #1:

The title says that delignification, but actually it appears to me that depolymerization reaction. Also, the results presented are not clear to some reproduced at other places. Also, need to rewrite many parts of the manuscript. Need to explain what happened to lignin.

The preparation and application of CNFs have attracted wide attention in academia and industry. At present, CNFs are mainly prepared by pretreatment and mechanical treatment. In fact, most of the lignin and hemicellulose from lignocellulosic biomass are removed during pretreatment and the remaining solid rich in cellulose are obtained. The process of lignin removal is often referred to as delignification in pulp and paper making. The title has been changed for the benefit of the reader. The main content of this study is the effect of *p*-TsOH pretreatment on the main components of bagasse. The ultimate goal is to efficiently remove lignin and hemicellulose while retaining the most cellulose. High performance CNFs were prepared to utilize cellulose - rich solids. So the new title is “Effect of *p*-TsOH pretreatment on separation of bagasse components and preparation of nanocellulose filaments”. The unclear descriptions in the manuscript have been modified and improved. In this study, the separation of bagasse components by *p*-TsOH pretreatment was investigated. The effects of *p*-TsOH dosage, reaction temperature, and reaction time on cellulose, hemicellulose and lignin contents were studied. CNFs were produced from the pretreated bagasse. The results show that the *p*-TsOH pretreatment is an efficient separation technique. It is also one of the effective pretreatment methods to prepare CNFs. The main contents of this study were the content change of major components of bagasse and the characterization of

CNFs prepared. The technical quality of the research reported is valid and appropriate.

At present, we are studying the structural changes of lignin during *p*-TsOH pretreatment. Thank you very much for your valuable comments. It improves the quality of the manuscript. I hope the author's reply can meet your requirements.

Reviewer #2:

A) In this manuscript, were LCNFs and CNFs prepared in the same way?

In fact, most of the lignin and hemicellulose from bagasse were removed by pretreatment with *p*-TsOH. But the cellulose was also damaged in the process, and some lignin remains in the remaining raw material. Their impact on the preparation of CNFs needs to be assessed. In the same way, pretreated bagasse and bleached bagasse were used to prepare CNFs and the control CNFs, respectively. Thus, the application value of *p*-TsOH pretreatment in the preparation of CNFs was analyzed.

B) It is suggested that Fig. 4. and Fig. 5. should be combined together to facilitate the reader's observation.

It has been modified as suggested. Fig. 4. and Fig. 5 have been merged. The manuscript has been revised and marked accordingly.

C) It is recommended to unify the font, such as “°C” in the first paragraph of Section 4.2.

The font has been unified in revised version.

D) Please specify the reason why 80% *p*-toluenesulfonic acid (*p*-TsOH) is the optimal condition for the LCNFs production.

High performance CNFs need to be produced by *p*-TsOH pretreatment. This means that the lignin and hemicellulose content in the pretreatment samples should be reduced as much as possible, while the cellulose should be protected during the process. The results showed that the content of lignin and hemicellulose were significantly reduced with the increase of *p*-TsOH concentration. The concentration of *p*-TsOH concentration had little effect on cellulose content. The reduction of lignin and hemicellulose was effective when the concentration of *p*-TsOH exceeds 80%. It also causes waste of

chemicals and higher disposal costs. Therefore, 80% *p*-TsOH is the optimal condition for the LCNFs production.

E) Are the three factors, the concentration of *p*-TsOH, temperature and time, worked independently? Please specify the reason why they were studied separately.

The effects of the three factors on the reaction are not independent. The effects of *p*-TsOH dosage, pretreatment temperature and pretreatment time on bagasse components were analyzed by single factor experiment. The effects of *p*-TsOH on the main components of bagasse were studied at fixed temperature and time. Then the effect of pretreatment temperature on the main components of bagasse was studied under the optimal *p*-TsOH dosage. The effect of pretreatment time on the major components of bagasse was analyzed at optimal dosage and temperature. The optimal process of *p*-TsOH pretreatment was obtained. These three elements also interact in the process of single factor analysis. In the future, their interactions will be studied by response surface design. The process conditions will be further optimized.

F) The images of the LCNFs and CNFs were highly recommended to be included in the manuscript.

Figure 4 shows the LCNFs and CNFs images.

As a whole, issues the reviewers suggested are very pertinent, which are very helpful to modified my entire paper and thank you very much again.